# Implementing primary healthcare-based measurement, advice and treatment for heavy drinking and comorbid depression at the municipal level in three Latin American countries: final protocol for a quasiexperimental study (SCALA study)

Eva Jané-Llopis [1,2,3] Peter Anderson [2,4] Marina Piazza,[5] Amy O'Donnell [4] Antoni Gual,[6,7,8] Bernd Schulte,[9] Augusto Pérez Gómez,[10] Hein de Vries,[2] Guillermina Natera Rey,[11] Daša Kokole,[2] Ines V Bustamante,[5] Fleur Braddick,[6] Juliana Mejía Trujillo,[10] Adriana Solovei,[2] Alexandra Pérez De León,[11] Eileen FS Kaner,[4] Silvia Matrai,[6] Jakob Manthey [12] Liesbeth Mercken,[2] Hugo López-Pelayo,[6,7,8] Gillian Rowlands,[4] Christiane Schmidt,[9] Jürgen Rehm[3,12,13,14,15]

For numbered affiliations see end of article.

**Correspondence to**
Eva Jané-Llopis;
eva.jane@esade.edu

## INTRODUCTION

This paper outlines the protocol for a quasi-experimental study[1] to test the implementation of primary healthcare (PHC)-based measurement, advice and treatment for heavy drinking and comorbid depression at the municipal level in three Latin American countries, Colombia, Mexico and Peru (Scale-up of Prevention and Management of Alcohol Use Disorders and Comorbid Depression in Latin America (SCALA) study).

Heavy drinking is a cause of considerable disability, morbidity and mortality.[2] Heavy drinking is a causal factor for some communicable diseases (including TB and HIV/AIDS), for many non-communicable diseases (NCDs, including cancers, cardiovascular diseases and gastrointestinal diseases) and for many mental and behavioural disorders, including depression, dementias and suicide.[3 4]

In PHC settings, two-fifths of people with heavy drinking have depression, with risks of incident depression higher for heavier as opposed to lighter drinkers.[5] In addition to its role in the aetiology of depression, heavy drinking is associated with worsening the depression course, including suicide risk, impaired social functioning and impaired healthcare utilisation.[6]

## Strengths and limitations of study

► Uses a theory-based approach to tailor clinical materials and training programmes, creating city-based Community Advisory Boards, and user-based user panels to ensure that tailoring matches user needs, municipal services and coproduction of health.

► Tests the added value of embedding and implementing primary healthcare activity within municipal-based adoption mechanisms and support systems, and community-based communication campaigns.

► Has a longer time frame (18 months) than is traditionally used in implementation studies, to assess longer term impacts.

► Gives considerable emphasis to process evaluation, developing logic models to document the fidelity of all implementation strategies, and to identify, the drivers and barriers and facilitators to successful implementation and scale-up.

► Due to municipal-based political and technical considerations, we are unable to randomise the involved municipal areas. We adopt a quasiexperimental design, optimising comparator municipal areas for confounding, and by using propensity score matching.

Heavy drinking is also a major contributor to global health inequalities, with alcohol-related harm aggravated by lower socioeconomic status[7] and extending beyond the individual

drinker to families, communities, health systems and the wider economy. Tackling the multiple individual and societal level harms caused by heavy drinking is essential for achieving global targets of reducing deaths from NCDs by 25% between 2010 and 2025,[8] more so as risk of exposure to harmful use of alcohol increases with increasing socioeconomic status.[9] In line with tackling harm due to lower socioeconomic status, United Nations Sustainable Development Goals include Target 3.5, to strengthen the prevention and treatment of harmful use of alcohol, with two proposed indicators: coverage of treatment interventions (pharmacological, psychosocial and rehabilitation and aftercare services) for harmful use of alcohol; and per capita alcohol consumption.[10 11]

Countries in Latin America have the highest alcohol-attributable disease burden after Eastern Europe and sub-Saharan Africa, with particularly high risks in alcohol-attributable traffic injury including violence.[12] The burden of alcohol-attributable diseases in Latin America lead to marked economic costs, with numerous calls to implement effective and cost-effective policies.[13]

A robust and extensive body of literature demonstrates the range of evidence-based strategies that can be implemented to reduce heavy drinking in healthcare settings.[14] Questionnaire-based measurement and brief advice programmes delivered in PHC are effective[15] and cost-effective[16 17] in reducing heavy drinking. In addition to brief advice, treatment for heavy drinking includes cognitive behavioural therapy and pharmacotherapy, both of which are found to be effective in reducing heavy drinking.[18] Were the proportion of eligible patients receiving advice and treatment for heavy drinking to increase to 30% of eligible patients, the prevalence of harmful use of alcohol could decrease by between 10% and 15% across OECD (Organisation for Economic Co-operation) and member countries.[19] However, to date, measurement and brief advice and treatment programmes have failed to achieve widespread take-up.[19]

Two systematic reviews[20 21] and two multicountry studies[22–24] have demonstrated that the proportion of PHC patients whose alcohol consumption is measured, and of heavy drinking patients given advice can be increased by providing training and support to PHC providers, although from very low baseline levels, and with effects not generally sustained over the longer term. Moreover, while there has been some previous research in countries of Latin America,[25–30] most implementation work to date has been undertaken in high-income countries. The SCALA study will build on previous evidence[31] to fast-track scale-up research and practice in Latin American PHC settings.

Out of a range of implementation frameworks that include a sequential approach for scale-up, and that provide practical guidance for how to work with organisations, health systems and communities to implement and scale-up best practices,[32–39] we adopt the Institute for Healthcare Improvement's Framework for going to Full Scale, which identifies adoption mechanisms and support systems for use across sequential steps, and describes the implementation methods that can be used at each step.[40]

SCALA seeks to address three specific barriers to sustained implementation of PHC-based measurement, advice and treatment for heavy drinking. The first barrier recognises that most PHC-based programmes focus on providers alone, whereas successful implementation of health interventions within complex health system demands addressing a range of underlying structural and support systems.[40] Phase IV of the WHO study on the identification and management of alcohol-related problems in primary care concluded that embedding PHC-based measurement and brief advice programmes within the frame of supportive community and municipal environments might lead to improved outcomes,[41] although this has never been formally evaluated. Similar conclusions were reached by the European Optimising Delivery of Healthcare Interventions (ODHIN) study[42] and the US-based Substance Abuse and Mental Health Services Administration Screening, Brief Intervention and Referral to Treatment initiative.[43–45]

The second barrier is that standard cut-off points for the frequently used alcohol measurement instrument, Alcohol Use Disorders Identification Test, 3-item consumption version (AUDIT-C)[46] (commonly a score of five for both men and women, or five for men and four for women) to trigger advice are too low,[47] being equivalent to an average daily alcohol consumption of about 20 g of alcohol (around two standard drinks) or less.[48] Practitioners may well find it problematic to give advice at such levels, which would also have huge time implications, with one in three or four patients being eligible for advice in many countries, under this criterion.[24 49] We have argued to adopt similar models to blood pressure, where cut-off points for managing raised blood pressure are often determined by levels of blood pressure at which treatment has shown to be effective.[50 51] Similarly, cut-off points for brief advice could be the baseline levels of alcohol consumption found in the randomised controlled trials that have investigated the effectiveness of PHC-delivered brief advice. In the first Cochrane review of the topic that focused on PHC, mean baseline levels were 313 g of alcohol per week,[52] equivalent to an AUDIT-C cut-off of 8.[48]

The third and final barrier concerns the cost of implementing measurement and brief-advice for heavy drinking in PHC setting. Although, alcohol advice and treatment programmes can lead to substantial reductions in healthcare costs,[16] freeing considerable numbers of working age people from alcohol-related diseases,[19] their initial implementation can require a significant time-commitment on the part of providers, in terms of both initial training requirements and the time taken to deliver advice in routine practice. The largest part of the costs of implementing measurement and brief advice for heavy drinking in PHC settings are directly caused by the time spent by the healthcare providers delivering this intervention.[53] Moreover, this large amount of time

is experienced by healthcare providers as an important barrier to deliver routine measurement and brief advice to their patients.[54] As evidence suggests that shorter sessions of brief advice are not less effective compared with longer sessions,[52 55 56] it seems that reducing the time spent by healthcare professionals in preparing for these sessions could be a viable strategy to increase the overall adoption and implementation of alcohol measurement and brief advice at PHC level.

Given the strong comorbidity between heavy drinking and depression, our protocol includes screening for depression for those patients identified as heavy drinkers, with appropriate referral or PHC support for treatment.[57–59]

In the SCALA study, we implement three interventions (independent variables) for the PHCU:

1. Intensity of clinical package and training (standard, vs short, vs none).
2. Training of providers (present, vs absent).
3. Community integration and support (municipal action present, vs absent).

The main outcome (dependent variable) is the cumulative proportion of the adult (aged 18+ years) population registered with the PHCU that has their alcohol consumption measured within the 18-month implementation test period (defined as coverage). Three hypotheses are to be tested.

### Hypothesis 1

Municipal action leads to more sustainable coverage. After 18 months, the difference in coverage between municipal action present and municipal action absent for those PHCU that receive training is larger than after 12 months.

### Hypothesis 2

In the absence of municipal action, PHCU that have received training obtain higher coverage than PHCU that do not receive training.

### Hypothesis 3

In the presence of municipal action, the short clinical package and short training do not lead to less measurement coverage than the standard clinical package and standard training.

## METHODS AND ANALYSIS

The study is a quasiexperimental design,[1] comparing changes in measurement and assessment for alcohol consumption and comorbid depression, and, if needed, advice and/or referral for treatment between PHCUs in intervention municipal areas and PHCUs in similar control municipal areas. In 2017, prior to a grant application, we published a preprotocol for a three-country study to test the scale-up of PHC-based programmes to identify and manage the harmful use of alcohol and comorbid depression.[60] Since the application, and during the grant

negotiation and planning phase, the design of the study has changed considerably, essentially moving from a two-arm design to a four-arm design, and changing the primary outcome measure to the proportion of the adult population registered with a PHCU that has their alcohol consumption measured, online supplementary file S1, online supplementary box S1. With all changes approved by the concerned ethics committee, this paper outlines the final protocol for a quasiexperimental study to test the implementation of PHC-based measurement, advice and treatment for heavy drinking and comorbid depression at the community level in three Latin American countries, Colombia, Mexico and Peru (SCALA study).

Intervention municipal areas are investigator-selected from Bogotá (Colombia), Mexico City (Mexico) and Callao—Lima (Peru). Control municipal areas are investigator-selected in the same cities, on the basis of comparability with the intervention municipal area in terms of socioeconomic and other characteristics which impact on drinking, healthcare and survival, comparable community mental health services and sufficient geographical separation to minimise spill over effects from the intervention municipal area. Randomised selection of the municipal areas was not feasible due to organisational limitations. Municipal areas are chosen as a scalable implementation unit at mesosystem level that can be replicated as the intervention is scaled-up,[40] given their jurisdictional responsibilities for prevention and healthcare services.

Within each intervention municipal area, a local Community Advisory Board (CAB) is created of key stakeholders, including representatives of local and regional government, directors of PHC services, non-governmental organisations active in providing counselling and treatment services for alcohol and mental health, academic experts and local media. The CABs meet regularly during the course of the study, giving advice on tailoring materials for local use, giving advice on adoption mechanisms, support systems and communication campaigns to support the action and preparing for sustainability and scale-up at the end of the action.

The units of allocation and analysis, that is, study participants, are 54 PHCUs and the providers working in them. Within each PHCU, eligible providers include any fully trained healthcare provider working in the PHCU and involved in medical and/or preventive care. Within each PHCU, individual providers decide themselves whether or not to participate in the study; those who do sign an informed consent for their participation. Based on the five-country ODHIN study, we estimate that approximately two-fifths of providers will consent to join the study.[61] The overall study design is summarised in figure 1. Fifty-four PHCU are invited to join the study until 27 are achieved within each of the two municipal areas (intervention and control) across the three countries (nine per municipal area within each of the three countries).

Within each intervention municipal area, a user panel (UP) is created of providers and patients drawn from

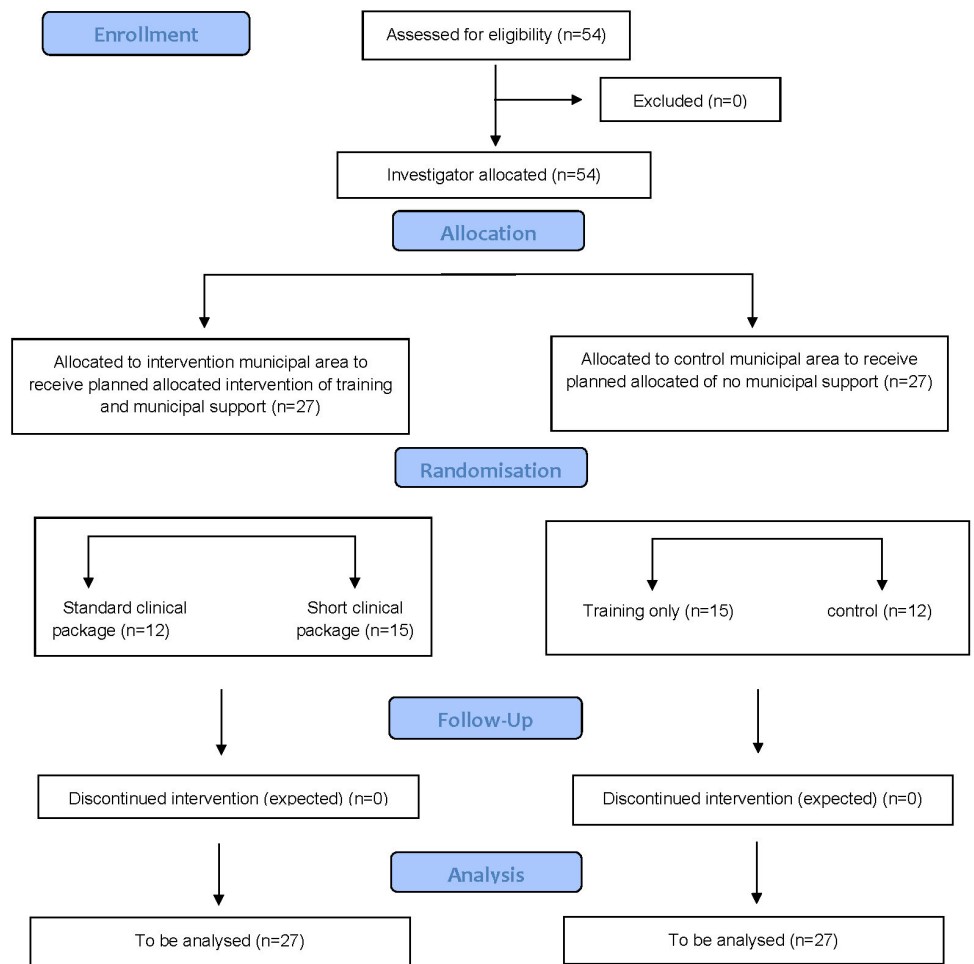

**Figure 1** Study flow diagram.

the PHC centres to advise on the tailoring of patient and provider materials and on provider training programmes.

For the first 6 months of the 18-month implementation and test period, a four-arm design is adopted, figure 2. Within the comparator municipal area, 12 PHCUs out of the 27 are randomly allocated to control (Arm 1), and 15 are allocated to receive short training to implement a short clinical package (Arm 2). Within the intervention municipal area, in which all 27 PHCU receive municipal

action, 15 PHCUs are randomly allocated to receive short training to implement a short clinical package (Arm 3), and 12 PHCUs are allocated to receive standard training to implement a standard clinical package (Arm 4). Random allocation was undertaken using Excel random number generator.

The clinical package comprises measurement instruments, patient information and advice material and provider guidance material, with the differences between

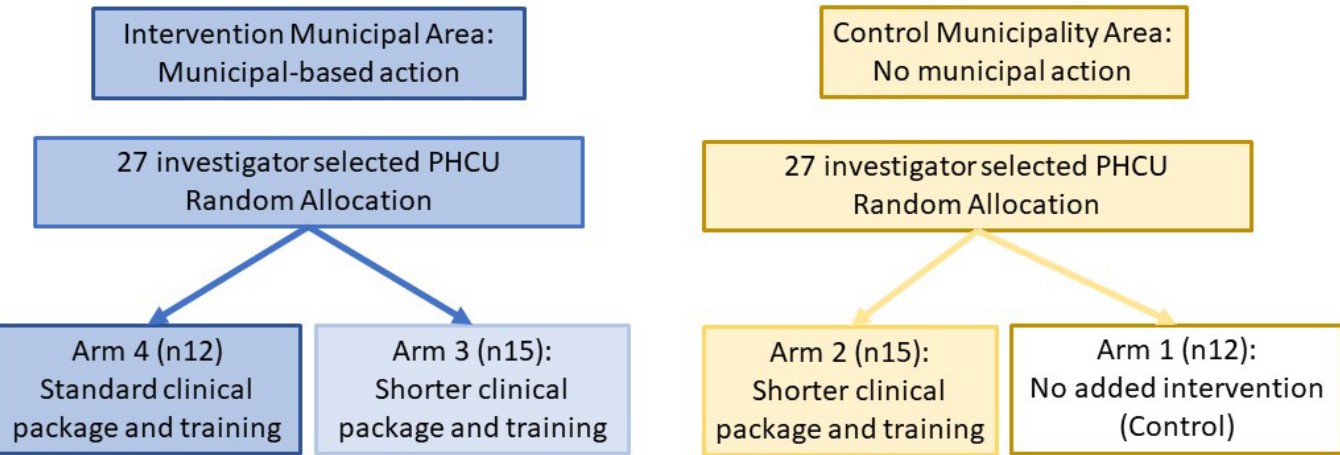

**Figure 2** Study design for the first 6 months of the 18-month implementation period. PHCU, primary healthcare units.

the standard and short clinical materials are described in online supplementary file S1, online supplementary table S1, with references. Online supplementary file S1, online supplementary table S1 also lists the material used in control Arm 1. The standard material is essentially that used in common clinical practice[60] and the short version a simplified version deliverable in practice during a short period of time. The packages include measurement instruments and patient advice material for comorbid depression implemented with patients with an AUDIT-C score of 8+. Online supplementary file S1, online supplementary table S1 summarises the differences between the standard and short versions of the training programme.

The standard and short care pathways that are implemented are summarised in online supplementary file S1, online supplementary figures S1 and S2.

Essentially, in all arms, PHC providers are asked to measure the alcohol consumption of all adult patients who consult for whatever reason using AUDIT-C. The three AUDIT-C questions are included in a paper tally sheet completed by the provider, in which the providers document the outcome of the consultation (advice given, patient referred, etc). The local researchers visit each PHCU on a 2–4 weekly basis to collect completed tally sheets and deliver new tally sheets as required. The local researchers collect information on the total number of adult patients (aged 18+ years) registered with each PHCU and the monthly number of total adult consultations with each provider. Patients who score <8 with AUDIT-C are given a patient information leaflet. Patients who score 8+ with AUDIT-C are assessed and manged as appropriate for depression, and are advised to reduce their alcohol consumption, unless there are clinical indications for referral. Arm 4 differs from Arm 3 in having a lengthier assessment, if indicated, and a longer session of advice giving.

By Month 6, Hypothesis 3, that is, non-superiority of Arm 4 (standard package with municipal action and standard training) over Arm 3 (short package with municipal action and short training) will be tested. In the presence of clinical equivalence of a relative difference of the primary outcome, that is, the cumulative coverage of patients whose alcohol consumption is measured, of less than 10%, Arm 4 will be replaced by Arm 3 from month 8 onwards, figure 3.

The municipal integration and support inputs to Arms 3 and 4 within the intervention municipal area are summarised in online supplementary file S1, online supplementary table S2, with references. Municipal integration and support comprises:
1. Creation of local CAB of local stakeholders to advise on tailoring of materials, support local implementation and review drivers of successful action.
2. Appointment of local project champion to advocate for successful implementation of programmes.
3. Implementation of five evidence-based adoption mechanisms.
4. Implementation of five evidence-based support systems.
5. Implementation of community-based communication campaigns.

### Tailoring

The CABs and UPs review and tailor relevant materials of the clinical package and training courses and of the municipal integration and support inputs within the seven domains of: (1) local and national guideline factors; (2) individual healthcare provider factors; (3) patient factors; (4) interactions between different professional groups; (5) incentives and resources; (6) capacity for organisational change and (7) social, political and legal factors.[62–64]

The study timetable is summarised in figure 4. The data management plan, as submitted to the European Commission, is available as online supplementary file S2.

**Figure 3** Study design from month 8 onwards, assuming no superiority of Arm 4 over Arm 3 during first 6 months of implementation. PHCU, primary healthcare units.

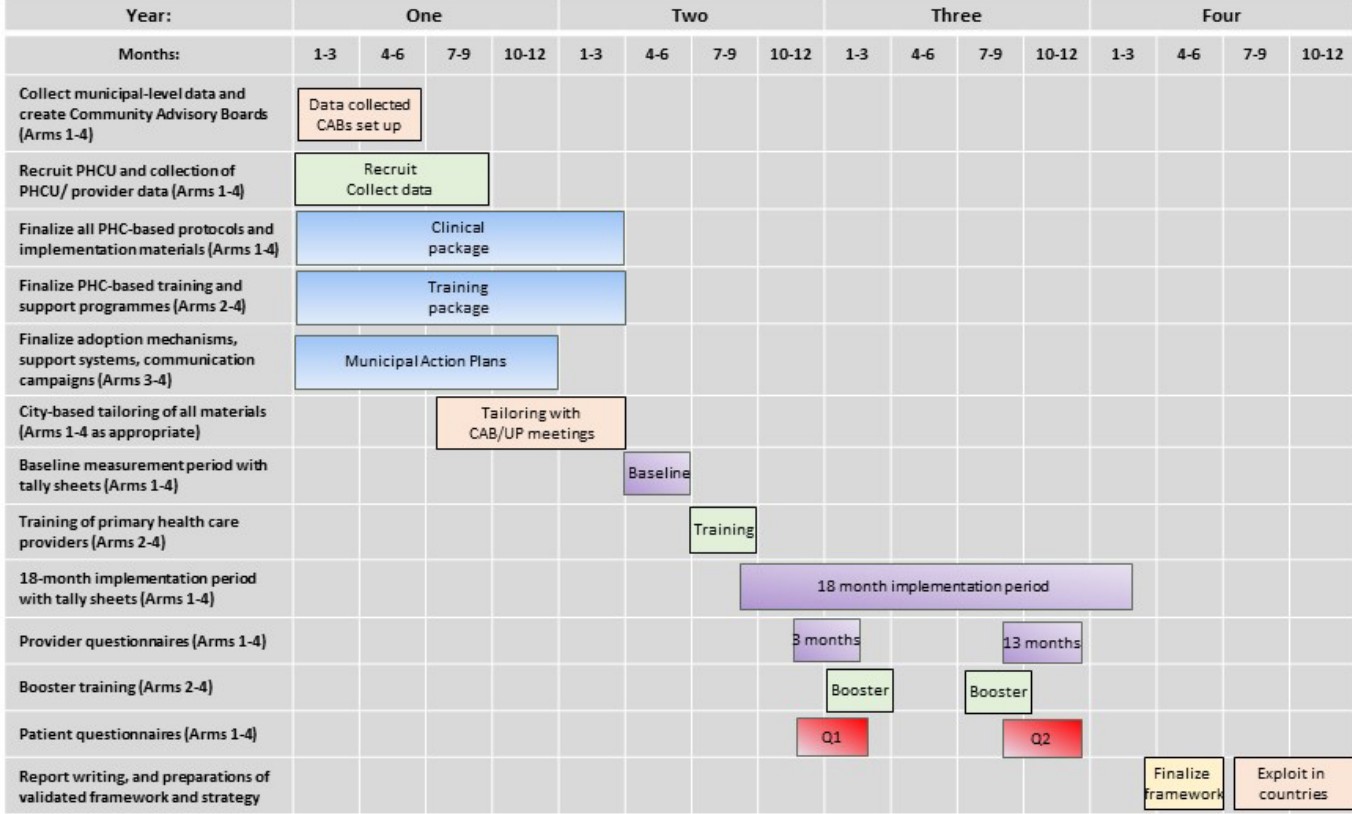

**Figure 4** Study timetable.

## Data collection and instruments

### During set-up phase for Arms 1–4

*Municipal level information*

At the level of the municipal area (or, when not available, at whole city, regional or country level), the following information will be collected from routinely available data on sociodemographic factors, alcohol and mental health data, health system structures, quality of life, sustainable governance and values, online supplementary file S1, online supplementary table S3.

*PHCU and provider level information*

All contacted PHCU, including those who did and did not agree to be part of the study, will provide information on the following:

▶ Numbers of registered patients, divided into age 0 to 17 years and 18+ years.
▶ Numbers and professions of provider staff (including physicians, nurses, nurse technicians, midwifes, psychologists, social workers and others).

At recruitment, PHC providers will provide data on their:

▶ Age.
▶ Gender.
▶ Profession (doctor, nurse, practice assistant, etc).
▶ Time worked in the PHC.
▶ Data on their attitudes and experiences to working with patients with heavy drinking and comorbid

depression (online supplementary file S1, online supplementary table S4).

Since we are unable to randomise the municipal areas involved, we will use propensity score matching (PSM) based on data collected at the level of the municipal area and the PHCU, to take into account potential confounding variables between control and intervention municipal areas, and minimise bias on account of these.

### During 1-month baseline measurement period for Arms 1–4

*Provider-based measurement and assessment of alcohol consumption and comorbid depression and record of advice and treatment given (tally sheets)*

Based on the validated methodology of the ODHIN project,[22 24] PHC providers will be asked to document activity by completing anonymous paper tally sheets that record eligible patients' (aged 18+ years) AUDIT-C scores,[65] and, if administered (as documented in online supplementary file S1, online supplementary table S1), AUDIT-10,[66] Patient Health Questionnaire (mental disorders), 2-item version (PHQ-2)[67] and PHQ-9[68] scores, and the advice or treatment given to each patient. The tally sheets will record the age, sex and educational level of the patient, the latter as a proxy measure of socioeconomic status. PHCUs will return data on the number of adult (aged 18+ years) consultations per provider for the 1-month baseline measurement period. Tally sheets will be delivered to the PHCU to be distributed to the

participating providers at the beginning of the 1-month baseline measurement period and collected at the end of the period, with no other contact during the period.

### During training prior to implementation for Arms 2–4

Providers will complete a short questionnaire after the initial training sessions. The questionnaires, which are adapted based on specific training contents (standard or short package), will assess the participants' experience of the training, measuring satisfaction with the components of the training aspects, as well as their perceived utility. Two measures included in the main provider questionnaires, Short Alcohol and Alcohol Problems Perception questionnaire (SAAPPQ)[69] and self-efficacy,[70] will be included in order to assess the specific impact of the training, independent of the effect of the implementation of the intervention.

### During 18-month implementation period for Arms 1–4

*Provider-based measurement and assessment of alcohol consumption and comorbid depression and record of advice and treatment given (tally sheets)*

The same mechanism, for tally sheets used during the baseline measurement period will continue for each calendar month of the 18-month implementation period. Tally sheets will be delivered monthly to each PHCU to distribute to participating providers. Completed tally sheets will be collected at the end of each month. Following training in Arms 2–4, and municipal support in Arms 3–4, each provider determines use and completion of the tally sheets, with no additional prompting. Monthly data will be collected and reported with accumulation of coverage over time. Formal reporting will be undertaken at baseline, and for coverage achieved by month 12 and by month 18 of the 18 month implementation and test period. Tally sheets will include an identifying code of the provider, PHCU, country and study arm, but no identifying code of the patient. Data will be extracted and sent to the project's data warehouse at Technical University Dresden on a monthly basis.

### Extended tally sheets

As part of quality control, in all four arms at two time points, during the 18-month implementation and test period (months 3 and 15), providers will complete extended tally sheets on two separate days in each month. The extended tally sheets will include an identifying code of the provider but no identifying code of the patient. The extended tally sheet will include additional information from the patient on alcohol knowledge,[71] social norms[72] and health literacy[73] applied to alcohol, as it informs the content of advice given; and, additional information from the provider on contextual characteristics that informed their advice giving. The extended tally sheets will include a consent form for the patient and self-completed additional questions for the patient to complete, once the consultation has ended.

### Self-completed additional questions by patient

On two separate days, during months 3 and 13, coinciding with and following the consultation with the provider using the extended tally sheet, patients who are able to read and write will be invited to give consent to self-complete additional questions to the extended tally sheet in the waiting room before leaving the PHCU, handing the completed tally sheet and questions to a researcher in attendance. No patient identifying information will be included in the questionnaires. Six domains, serving as quality control, will be included:
1. AUDIT-C.[65]
2. PHQ-2.[67]
3. Experiences of the consultation.
4. Views on being asked about alcohol consumption.
5. Health literacy[73] as it applies to alcohol.
6. Exposure to communication and media campaigns on alcohol.

On each day, 270 patient questionnaires will be collected across all PHCUs, with up to 1080 (540 during each of months 3 and 13) questionnaires completed in total across the 4 days.

### Provider-based attitudes and experiences

At two time points during the 18-month implementation period (months 3 and 13), providers will provide data on their attitudes and experiences to working with patients with heavy drinking and comorbid depression, online supplementary file S1, online supplementary file S2.

Providers will complete a short questionnaire after each of the booster training sessions that they attended (at months 4 and 8). The specific content, number and timing of the training-related questionnaires will depend on the study arm: Arm 2 and 3 participants will fill in one questionnaire after the booster session; while Arm 4 participants will fill in two after each of the two booster sessions.

### Observations

The training sessions with the PHC providers, and the meetings of the CABs will be observed by a neutral observer in order to take note of additional possible barriers in the implementation of the protocol that emerge through the training sessions and meetings. Participant responsiveness will also be observed.

### Economic data for return-of-investment analyses

Within SCALA, we will conduct return-on-investment (RoI) analyses, by assessing for each EURO invested in scaling up delivery of screening and brief interventions in PHC in Columbia, Mexico and Peru, how many EUROs will be saved by reductions in future healthcare utilisation. The return of investment will be defined as the (return on investment=(gain from investment–cost of investment)/cost of investment). For details on the data required for RoI analyses, online supplementary file S1, online supplementary table S5.

For the RoI analyses, the effects of increased coverage of alcohol brief advice among PHC patients will be modelled

using effect sizes from previous meta-analyses.[52 74] To translate the reduced intake of alcohol into health gains, we will calculate alcohol-attributable fractions for major disease and injury categories. These fractions will then be applied to the cost data outlined in online supplementary file S1, online supplementary table S5 to estimate the alcohol-attributable costs per disease category.

## Process evaluation

As the intervention is embedded in a complex system involving actions and actors at different levels (individual, organisational and municipal), a thorough process evaluation will be carried out to complement and better understand the outcomes. Through the process evaluation, the implementation with its fidelity and adaptation will be assessed, along with the drivers of scale-up and contextual factors influencing the implementation, the drivers and the outcomes. This will be achieved in four blocks: driver diagram creation; barriers and facilitators analysis; assessment of implementation, mechanisms of impact and context and further contextual and policy analysis.

### Key informant interviews

A number of individual or group interviews will be undertaken throughout the project with key stakeholders—providers, UP members, CAB members, municipal and PHC-based clinical leaders, project partners and any other people involved in the implementation of the SCALA project. Depending on the stakeholder and their involvement in the project, the topics of the interviews will cover topics such as the necessary adaptation to the protocol; the experience of implementing the programme in PHC practice and the perception of the municipal support and the community campaigns.

### Driver diagrams

Driver diagrams[75] will be used in order to describe the intervention and its causal assumptions, providing the theory of change through displaying what contributes to intervention aim and what are the relationships between primary drivers, secondary drivers and specific change ideas/activities. The initial general driver diagram, online supplementary file S1, online supplementary figure S3, will be modified based on local contexts and adapted throughout the duration of the project in order to understand how scale up varies in the different cities.

### Barriers and facilitators assessment

Factors influencing the implementation of the SCALA protocol will be assessed before the implementation, as well as monitored throughout. The anticipated barriers and facilitators to implementation will be assessed through development of evaluation tool based on literature review[76–78] and implementation framework,[62] with subsequent refinement and adaptation to the local context through focus group discussions and workshops with the CABs. The aim of the tool is to identify the barriers that would have to be addressed and monitored throughout implementation and the facilitators that

would incentivise and engage providers and the primary healthcare unit (PHCU) managers in uptake and scaling up of the SCALA protocol. The experienced barriers and facilitators will be further monitored through meeting observations, provider questionnaires and interviews, as well as interviews with other involved stakeholders (eg, CAB members, PHCU managers).

### Implementation, mechanisms of impact and context

The factors influencing the progress from scale-up to outcomes will be identified and documented based on UK Medical Research Council guidance,[79] analysing factors within five groups: (1) description of intervention and its causal assumptions, (2) implementation, (3) mechanisms of impact, (4) context and (5) outcomes. All aspects of the intervention will be taken into consideration: the intervention, intervention tailoring, training, training tailoring, as well as the municipal action, consisting of the CABs and the communication campaign, combining both quantitative and qualitative methods in order to obtain a comprehensive picture of the integration and interaction of included variables. A detailed description of the topics of interest and accompanied methods is presented in online supplementary file S1, online supplementary table S6.

The five groups will be assessed as follows:
1. Description of the intervention. The description of the intervention and its causal assumptions draws from the previously described driver diagram.
2. Implementation. Delivery of the training will be assessed though document analysis (reports from training), observation and self-reports from the trainers. Delivery of the intervention will be assessed through document analysis, interviews with patients and providers. The areas of focus will be fidelity, adaptation, dose and reach. Implementation of the CAB meetings and community action will be assessed mainly through document analysis, as well as key informant interviews.
3. Mechanisms of impact. The following three areas will be covered: participant responses to the intervention, mediators and unintended consequences. Mechanisms of impact of intervention delivery will be assessed through patient and providers' questionnaires. The patient interviews will focus on their responsiveness to the intervention, specifically looking at perceived acceptability. In order to evaluate participants' responses to the training, a post-training questionnaire examining satisfaction with the training and perceived utility of training sessions will be applied, triangulated with data from observation and trainers' self-report. Additionally, providers' self-efficacy will be tested as potential mechanism of impact that links the implementation to the outcomes. Mechanisms of impact of the CAB meetings and community action will be examined through key informant interviews and questionnaires. Specific focus will be placed on perceptions and mechanisms of actions of the communication campaign, examining its effect on attitudes and social norms of both providers and patients.

4. Context. Contextual factors that should be considered in order to better understand the success of the intervention will be assessed through meeting observation, document analysis and provider questionnaires, as well as stakeholder interviews, with the main focus primarily on individual and organisational level characteristics of the context. For the training evaluation, context will be assessed through observation and trainers' self-report. Context of municipal level actions will be assessed through key informant interviews. Additionally, contextual and policy factors on national and municipal levels will be assessed as described below.

5. Outcomes. The data collected through process evaluation will be combined with the outcomes and presented within the Reach, Effectiveness, Adoption, Implementation and Maintenance framework,[80–82] evaluating SCALA's impact across the dimensions of reach, effectiveness, adoption, implementation and maintenance.

## Contextual and policy factors

Based on methodology of Ysa et al,[83] contextual and policy factors on national and municipal level will be identified through document analysis and key informant interviews. The main variables considered for contextual analysis will be: (1) available data similar to that of the OECD better life initiative;[84] (2) Sustainable Governance Indicators[85] and (3) World Values Survey data.[86] For policy analysis, the information sought will be for a for alcohol policy-related strategies, action plans, legislation and evaluations, both on country and municipal levels. The existing contextual and policy factors will be mapped onto the test of the scale-up of the SCALA package to describe and identify those factors on national and municipal level that might influence going to full scale beyond the tested scalable units.

## Outcomes
### Primary outcome

The primary outcome will be the cumulative proportion of the number of adults (aged 18+ years) registered with the PHCU that have their alcohol consumption measured with a completed AUDIT-C instrument during the study period (coverage). The number of adults registered is provided by the administrative office of the PHCU and includes all adult patients covered by the PHCU, whether or not they consult during the 18-month implementation test period.

### Secondary outcomes

► Proportion of consulting patients who have their alcohol consumption measured by AUDIT-C: calculated as the number of adults who have their alcohol consumption measured by AUDIT-C divided by the total number of adults who consult the PHCU during the same time period per participating provider and per PHCU.

► At risk population receiving advice and/or treatment for heavy drinking: calculated as the number of adults with an AUDIT-C score of 8+ who receive brief advice

and/or referral for their heavy drinking divided by the total number of patients with an AUDIT-C score of 8+ per participating provider and per PHCU. Information will also be collected on the number of patients with an AUDIT-C score of <8 who receive brief advice and/or treatment for their heavy drinking.

► Proportion of patients with AUDIT-C score of 8+ who receive assessment for depression: calculated as the number of consulting adults with an AUDIT-C score of 8+ who complete PHQ-2 divided by the total number of patients with an AUDIT-C score of 8+ per participating provider and per PHCU.

► At risk population receiving advice and/or treatment for comorbid depression: calculated as the number of adults with a PHQ-2 score of 3+ who receive a patient leaflet and/or referral for their depression divided by the total number of patients with a PHQ-2 score of 3+ per participating provider and per PHCU.

► Provider attitudes: attitudes of the participating providers will be measured by the SAAPPQ.[65] The responses will be summed within the two scales of role security and therapeutic commitment. Individual missing values for any of the items in a domain will be assigned the mean value of the remaining items of the domain before summation.

## Statistical tests of key hypotheses
### Primary study goal

Multilevel regression analyses will be undertaken at 12 months' time of the implementation test period, using cumulative results at months 1 to 12, and at 18 months' time using cumulative results months 1 to 18. Both analyses will include covariates of country and results during baseline month, analysed at the levels of the PHCU by study arm, taking into consideration the hierarchical nature of the data. For any PHCU that drops out during the study, outcome values for subsequent measurement points will be set at the last value obtained.

### Hypothesis 1

Municipal action leads to more sustainable coverage among PHCU that receive training. We will compare results on primary outcome after 18 months with results after 12 months between Arm 3 versus Arm 2 via regression.

Dependent variables:
► For each PHCU, cumulative results of months 1–18 of number of patients whose alcohol consumption is measured with AUDIT-C per 1000 registered patients; and cumulative results of months 1–12 per 1000 registered patients.

Random effects:
► Country as random intercept (test for inclusion).

Independent variables:
► Proportion of consulting patients who have their alcohol consumption measured with a completed AUDIT-C instrument during the baseline measurement month.

► Condition:
  – Municipal action (yes vs no).
► Covariate:
  – Proportion of consulting patients who have their alcohol consumption measured with a completed AUDIT-C instrument during the baseline measurement month.

It is postulated that coverage for Arm 3 will be significantly higher than for Arm 2.

### Hypothesis 2

Training leads to higher coverage than no training. For both months 1–12 and months 1–18, compare cumulative coverage as per primary outcome between Arms 1 and 2 via multilevel regression analyses.
Dependent variable
► Cumulative results months 1–12, and cumulative results months 1–18 of number of patients whose alcohol consumption is measured with AUDIT-C per 1000 registered patients with PHCU.
Random effects:
► Country as random intercept (test for inclusion).
Independent variables:
► Condition:
  – Training (Arm 2 vs Arm 1).
► Covariate:
  – Proportion of consulting patients who have their alcohol consumption measured with a completed AUDIT-C instrument during the baseline measurement month.

It is postulated that coverage for Arm 2 will be significantly higher than for Arm 1.

### Hypotheses 3

In the presence of municipal action, the short clinical package and short training do not lead to less coverage than the standard clinical package and standard training. In the presence of clinical equivalence of a relative difference of cumulative coverage of patients screened by less than 10% by month 6, the difference between Arm 3 (all 15 PHCU across the three countries) and Arm 4 (all 12 PHCU across the three countries) will be assessed with regression analyses. If Arm 4 is not superior to Arm 3, both arms will be collapsed into Arm 3 (shorter package) from month 8 onwards.
Dependent variable:
► Cumulative results months 1 to 6 per 1000 patients.
Random effects:
► Country as random intercept (test for inclusion).
Independent variables:
► Condition:
  – Length of clinical package (longer=Arm 4 vs shorter=Arm 3).
► Covariate:
  – Proportion of consulting patients who have their alcohol consumption measured with a completed

AUDIT-C instrument during the baseline measurement month.

It is postulated that Arm 4 is not significantly superior to Arm 3.

### Sample size calculations for main hypothesis

As the outcome of the primary study goal is predicted to be Arm 3>Arm 2>Arm 1, we compared both Arm 2>Arm 1, and Arm 3>Arm 2.

Our power calculations are based on the following assumptions: given an average size of a PHCU of approximately 15 000 adults, with an average of 1500 new consultations per month, we expect a cumulative coverage after 12 months of 0.0325 of the registered adult population to have had their alcohol consumption measured in the control condition (Arm 1) (data extrapolated from month 3 and month 9 assessments of control group from ODHIN study;[22 24] Anderson, personal communication). For the short clinical package and short training (Arm 2), we expect this to increase to 0.075 (data extrapolated from month 3 and month 9 assessments of training group from ODHIN study;[22 24] Anderson, personal communication). Although the WHO Phase IV study predicts an additional beneficial impact of municipal support,[41] precise empirical data is not available—however, we consider an estimate for Arm 3, with municipal support, to be 0.15, a proportion that would need to be achieved to consider municipal support to be worthwhile. To detect the difference between Arm 2 and Arm 1, assuming a design effect of 15 PHCUs (clusters) across the three municipal areas in Arm 2, with 15 000 patients (items), and 12 PHCUs (clusters) in Arm 1, with 15 000 patients (items), with an ICC for PHCUs of 0.03 (data from ODHIN study;[22 24] Anderson, personal communication) we would have 82% power at a significance level of 5%.[87] For the difference between Arm 3 and Arm 2 (15 PHCUs/clusters in each arm), we would have 96.5% power.

### Patient and public involvement

Patients were not involved in the design of the study but are involved in the tailoring processes. Existing literature suggests that most patients find it acceptable for PHC providers to ask about their drinking using validated measurement instruments, and support the delivery of brief advice to those drinking above recommended levels.[88–96] However, the majority of the evidence to date draws on research conducted in Europe, and thus the findings are potentially less transferable to Latin American populations. In order to ensure the design and content of the intervention package, including related outcome measures, are appropriate for implementation in the target SCALA sites, we work closely with patients in each city to tailor patient materials. Within the intervention municipal areas in each of the three countries, UPs are created with representatives of patients from the PHC centres. As part of the tailoring process, people and patients within the UPs have the opportunity to comment on the materials and information designed for use by

patients. The results of the study will be disseminated directly to patients and the public through information made available via the PHCUs.

## DISCUSSION

The study has several features worth mentioning. It:

1. Uses a theory-based approach[62–64] to tailoring clinical materials and training programmes, creating city-based CAB, and user-based UPs to ensure that tailoring matches user needs, municipal services[97] and coproduction of health.[98]

2. Sets a higher cut-off score for AUDIT-C (8+) than is commonly used to trigger advice-giving, matching definitions of heavy drinking[99 100] and similar to baseline levels of alcohol consumption in PHC-based trials to reduce heavy drinking.[52] We set the same cut-offs for men and women, based on epidemiological evidence,[101] and to minimise unintended consequences of using different cut-offs for men and women.[102] We recognise the importance of comorbid depression by building in identification, management and referral mechanisms.[57–59]

3. Tests for non-superiority of implementing a standard measurement and 5 min brief advice intervention with 6 hours of training, compared with implementing a shorter 1 min brief advice intervention with 3 hours of training, taking into account that brief advice is as effective and cost-effective as more extended advice or treatment in reducing heavy drinking[55 103 104] and the need for very brief clinical and training programmes for time-constrained providers.

4. Tests the added value of embedding and implementing PHC activity within municipal-based adoption mechanisms and support systems,[40] and communication campaigns over and above training programmes solely directed to PHC providers.

5. Has a longer time frame (18 months) than is traditionally used in implementation studies,[105 106] to assess longer term impacts.

6. Gives considerable emphasis to process evaluation,[79] developing logic models to document the fidelity of all implementation strategies, and to identify, the drivers and barriers and facilitators to successful implementation and scale-up, and the political and economic contextual factors that might influence scale-up.

There are some limitations to the study design. A trial with random assignment of municipal areas is not feasible due to municipal-based political and technical considerations. As we are unable to randomise the involved municipal areas, we adopt a quasiexperimental design,[1] trying to optimise control municipal areas for confounding, and by using PSM. While full comparisons via randomisation, and thus establishment of causality, are not possible, together with the qualitative evaluation component of the study, we will be able to clearly identify the mechanisms which were crucial in leading to the outcomes. According to a recent seven-item checklist for classifying quasiexperimental studies for Cochrane reviews,[107] our approach is,

nevertheless, ranked as a strong design, online supplementary table S7.

Although our focus on embedding PHC activity within supportive municipal actions is hypothesised to increase measurement and brief activity over and above that previously demonstrated, such an approach also brings risks. Municipal and national governments change; and, thus health priorities may change. Although our approach minimises the need for extra resources (and in some jurisdictions, could be resource saving),[19] it is not resource free. Funding constraints could limit future scale-up and sustainability.

We have based our protocol adopted on a model of transdisciplinary research to promote sustainability. Such a model identifies, structures, analyses and deals with specific problems in a way that grasps the complexity of problems;[108] it takes into account the diversity of real-world and scientific perceptions of problems; and develops knowledge and practices that promote what is generally accepted to be the common good.[109] As such, we include municipalities and health systems as stakeholders to form explicitly orchestrated and managed ecosystems that cross-organisational boundaries. Municipal areas and health systems create an engagement platform that provides the necessary environment, including people and resources, for sustainability.

## ETHICS AND DISSEMINATION

This protocol outlines a quasiexperimental study[1] to test the extent to which embedding PHC-based measurement and brief advice activity within supportive municipal action leads to improved scale-up of an intervention package, with more patients having their alcohol consumption measured, and with heavy drinkers receiving subsequent appropriate advice and treatment. It is not envisaged that there will be any substantial protocol modifications during the course of the study. Any modification to the protocol will be described in all scientific publications.

All participating PHCUs and participating PHC providers sign an informed consent form for participation with the country-based research team. Selected patients at two separate time points sign an informed consent form with the country-based research team to provide additional anonymised information following a consultation with a PHC provider. The consent forms are included within Annexe Data Management Plan. All data collection, processing, and sharing procedures will adhere to national and international laws including the General Data Protection Regulation (EU Regulation 2016/679), as described within the Annexe Data Management Plan.

All materials are publicly available on the project website: https://www.scalaproject.eu/. According to the SCALA data management plan, by default, all quantitative data sets generated in the course of the SCALA study will be made openly available through the UK Data Service on publication of the results (http://www.data-archive.ac.

uk/). Prior to publication, all data will be formatted to meet UK Data Service requirements.

Ministries of Health at municipal and country levels are represented in the CAB created in each intervention municipality to facilitate scale-up at municipal and country levels, once the implementation strategy is validated. SCALA works closely with the Pan American Health Organization (PAHO), with the principal investigator form Mexico being a Collaborating Centre with PAHO, to facilitate scale-up at Latin American levels, once the implementation strategy is validated.

**Author affiliations**
¹ESADE Business School, Ramon Llull University, Barcelona, Catalunya, Spain
²Department of Health Promotion, Care and Public Health Research Institute (CAPHRI), Maastricht University, Maastricht, The Netherlands
³Institute for Mental Health Policy Research, Centre for Addiction and Mental Health, CAMH, Toronto, Ontario, Canada
⁴Population Health Sciences Institute, Newcastle University, Newcastle upon Tyne, UK
⁵Public Health and Administration, Universidad Peruana Cayetano Heredia, Lima, Peru
⁶Addiction Unit, Hospital Clínic de Barcelona, Barcelona, Catalonia, Spain
⁷Red de Trastornos Adictivos, Instituto Carlos III, Madrid, Spain
⁸Institut d'Investigacions Biomèdiques August Pi Sunyer (IDIBAPS), Universitat de Barcelona, Barcelona, Spain
⁹Center for Interdisciplinary Addiction Research (ZIS), Department of Psychiatry and Psychotherapy, University Medical Center Hamburg-Eppendorf, Hamburg, Hamburg, Germany
¹⁰Department of Research, Corporación Nuevos Rumbos, Bogota, Colombia
¹¹Dirección de Investigaciones Epidemiológicas y Psicosociales, Instituto Nacional de Psiquiatría Ramón de la Fuente Muñiz, Mexico, DF, Mexico
¹²Institute for Clinical Psychology and Psychotherapy, TU Dresden, Dresden, Germany
¹³Dalla Lana School of Public Health, University of Toronto, Toronto, Ontario, Canada
¹⁴Department of Psychiatry, University of Toronto, Toronto, Ontario, Canada
¹⁵Department of International Health Projects, Institute for Leadership and Health Management, I.M. Sechenov First Moscow State Medical University, Moscow, Russian Federation

**Contributors** EJL, PA, MP, AOD, AG, BS, APG, HdV, GNR, DK, IVB, FB, JMT, AS, APDL, EFSK, SM, JM, LM, HLP, GR, CS and JR contributed to the Grant Application, on which this protocol is based and adapted; commented on drafts of the manuscript and read and approved the final version. EJL drafted the first version of the paper, and revised the paper based on author's feedback and comments. PA prepared the paper and material for submission and undertook the submission process. PA undertook random allocation generation. APG and JMT assigned PHCU to arms in Colombia; GNR and APDL assigned PHCU to arms in Mexico; MP and IVB assigned PHCU to arms in Peru.

**Funding** The research leading to these results or outcomes has received funding from the European Horizon 2020 Programme for research, technological development and demonstration under Grant Agreement no. 778048—Scale-up of Prevention and Management of Alcohol Use Disorders and Comorbid Depression in Latin America (SCALA). Participant organisations in SCALA can be seen at: www.scalaproject.eu. The views expressed here reflect those of the authors only and the European Union is not liable for any use that may be made of the information contained therein. The Funder was not involved in the study design. The funder will not be involved in the collection, analysis, interpretation of data and preparations of any publication.

**Patient and public involvement** Patients and/or the public were involved in the design, or conduct, or reporting or dissemination plans of this research. Refer to the Methods section for further details.

**Patient consent for publication** Not required.

**Ethics approval** The Ethics Committee of the Technical University of Dresden gave final ethical approval for the SCALA project on 12 April 2019, EK90032018.

**Provenance and peer review** Not commissioned; externally peer reviewed.

**ORCID iDs**
Eva Jané-Llopis http://orcid.org/0000-0002-1739-3635
Peter Anderson http://orcid.org/0000-0003-4605-9828
Amy O'Donnell http://orcid.org/0000-0003-4071-9434
Jakob Manthey http://orcid.org/0000-0003-1231-3760

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
