## [Reviewer comments · BMJ Open]

ARTICLE DETAILS

TITLE (PROVISIONAL)	Implementing primary health care-based measurement, advice and treatment for heavy drinking and comorbid depression at the municipal level in three Latin American countries: final protocol for a quasi-experimental study (SCALA study)
AUTHORS	Jane Llopis, Eva; Anderson, Peter; Piazza, Marina; O'Donnell, Amy; Gual, Antoni; Schulte, Bernard; Perez, Augusto; de Vries, Hein; Natera Rey, Guillermina; Kokole, Dasa; Bustamante, Ines; Braddick, Fleur; Mejía Trujillo, Juliana; Solovei, Adriana; Pérez De León, Alexandra; Kaner, Eileen; Matrai, Silvia; Manthey, Jakob; Mercken, Liesbeth; López-Pelayo, Hugo; Rowlands, Gillian; Schmidt, CS; Rehm, Jürgen

VERSION 1 – REVIEW

REVIEWER	Hildi Hagedorn Center for Care Delivery & Outcomes Research, Minneapolis VA Health Care System USA
REVIEW RETURNED	23-Mar-2020

GENERAL COMMENTS	This protocol addresses a very pressing need to implement alcohol use screening, brief advice and referral to treatment into primary care settings. The described project is incredibly ambitious covering three countries and dozens of Primary Care Treatment Units. The design of the project is also commendable for incorporating implementation strategies that go beyond educating providers and simplifying tasks for them, e.g., utilizing community advisory boards to understand the context of implementation efforts, identify barriers and facilitators of implementation, and identify additional strategies to promote adoption, implementation and maintenance of the intervention. The extensive and detailed process evaluation plan will allow for collection of data necessary to understand variable results across PCTUs, municipalities and possibly even countries. With this ambitious and complex project comes the substantial challenge of presenting the research plan in a way that is understandable and accessible to the reader. My comments focus on areas of confusion that I encountered trying to understand the details of how the project will be carried out and suggestions for ways to possibly enhance the organization of the protocol paper. Major revisions: 1. It is not clear how many PCHUs will be involved in the study. Figure 1 suggested that there will be a total of 54. However, Figures 2 and 3 show an N of 9 per arm. I am wondering if there are multiple waves of PCHUs starting up? Maybe three waves of 18? But this is not clear from the paper.
---

	2. Clarifying the number of PCHUs involved as well as the timeline of when they will come on board would also help to understand the non-inferiority test of the short clinical package vs. the standard clinical package. I am assuming that this will be tested with the first 9 PCHUs enrolled in the intervention arm and then for future enrolled sites, the standard clinical package arm will be eliminated (if hypothesis of non-inferiority is confirmed) but again this is not clear from the paper. 3. The authors include a timeline for the project but this is buried in the supplemental attachments, not referenced in the paper, and overly detailed. I suggest the authors create a timeline specific for the protocol paper that shows just key elements such as when sites will be onboarded and when assessments will take place. This should be presented early on in the paper to orient the reader to the overall project. 4. As presented currently, the Data Collection and Instruments section is a list of instruments that are not organized by when the assessments will occur again making it difficult for the reader to gain a coherent understanding of what is occurring when during the project. I would suggest organizing the section by time, for example describing information collected prior to implementation, during provider training, during patient recruitment phase and post-implementation period. This can be tied to the timeline suggested above. 5. Key informant interviews and observations seem to be part of the Process Evaluation and so should be described under that section. 6. In the “Implementation, mechanisms of impact and context” section, the authors describe analyzing factors within five groups. The five groups are listed in the first paragraph of this section, then repeated under “The five groups will be assessed as follows:” and described in Table 6. Please, keep the order of the constructs consistent across all three of these and include all five constructs in Table 6. Currently Description of the Intervention and Outcomes are missing from the table. 7. There needs to be some additional clarity on how the patient tally sheets will be collected. Is it going to be a requirement that each provider use these sheets with every patient? Is this the case for all arms? Or is it up to each provider whether or not they use the tally sheet and for whom? Is it providers that agree to be enrolled in the study that are using the tally sheets and those who decline participation are not? I ask these questions because the primary outcome is the portion of patients who are assessed with the AUDIT-C which is part of the tally sheet so it is very important for the reader to understand the parameters around how that data is collected. Minor issues: 1. Many acronyms throughout are not spelled out on first use and may not be familiar to all readers, e.g., PHC, NCD, OECD, ODHIN, SBIRT
--	--

REVIEWER	Eric Hawkins VA Puget Sound HCS Department of Psychiatry and Behavioral Sciences, University of Washington
REVIEW RETURNED	11-Apr-2020

GENERAL COMMENTS	The authors describe a protocol for a quasi-experimental, four-arm design study designed to implement a primary care-based prevention and management of heavy drinking intervention among
---

municipal areas of three countries. A total of 54 primary care-based clinics will be recruited from municipal areas across Columbia, Mexico and Peru. One-half of the primary care clinics will be recruited from areas with no municipal support for the project. Participants will be primary care-based clinics and providers in those clinics. Data related to primary outcomes will be collected monthly for the 18-month duration of implementation activities and will include screening and intervention data. Data will also be collected from municipal areas, providers and patients. Multilevel regression analyses will be used to evaluate primary outcomes at 12 and 18 months. Analyses will also include return-on-investment, process evaluation as well as an analysis of contextual, financial and political-economy influencing factors.

Overall, this manuscript is well written and has several strengths. The authors deserve praise for taking on such an important and large study that could improve the field's understanding of the factors potentially influence implementation of screening and treatment interventions in large, geographically and culturally diverse health care systems. However, there were several reporting features that could be clarified to improve the clarity of the protocol.

Abstract:

- 1) Include the number of primary care-based clinics that are proposed to be enrolled in and the aims of the study.
- 2) Briefly clarify the type and duration of training that primary care-based clinics will receive. For example, is it training on screening and managing heavy drinking?
- 3) Consider noting that the study is a four-arm (or three-arm) quasi-experimental study to clarify the original design and inform comparisons on primary outcome.
- 4) From the abstract and introduction, it is not clear how comorbid depression will be addressed in the study.

Introduction:

- 1) Several acronyms have not been defined (e.g., OECD, NCD, CAB).
- 2) Provide a reference for sentence in the introduction that reports "one in three or four patients being eligible for advice in many countries."
- 3) The protocol indicates three interventions, including intensity of clinical package and training, training of providers and community integration and support. However, the difference between the two interventions involving training is not clear. Also, consider modifying Hypothesis 2 to note that it is limited to comparator clinics without municipal support.
- 4) Although the rationale for including comorbid depression is provided, the focus and description of the procedures related to depression are limited.
- 5) Because the term "clinical package" is somewhat vague and was not formally defined in the narrative, consider replacing the terms "clinical package" with "alcohol screening and treatment procedures" and/or formally defining what clinical package means early in the protocol.

Methods:

- 6) Clarify on Figure 1 when the randomization procedure occurs. The narrative suggests the training and screening procedures are randomized. Additionally, the screening and training procedures

	on Figure 1 are not consistent with what is reported for comparator clinics on Figures 2 and 3. 7) Without information on the “clinical packages” presented early in the methods, it is difficult to fully understand the comparison between the standard and short clinical package and related trainings for Hypothesis 3. Essentially, what are the standard vs. short clinical packages and related trainings? For example, the information presented under the heading “Provider-based measurement and assessment of alcohol consumption...” is difficult to follow without defining the differences in the screening/treatment packages. 8) Clarify whether municipal and primary care based clinical leaders are considered key stakeholders for the purpose of key informant interviews. 9) Although referred to in the proposal, little information is provided about community advisory board and user panels, including how they are formed and the purpose they serve. 10) Given the abundance of statistical comparisons in a large sample, consider using an approach to correct for the number of comparisons. While reviewing the results on the Tables, there are several comparisons of means that are statistically significant but represent clinically insignificant differences. A higher p-value threshold might eliminate some of these differences, which would make it easier to present and highlight the results that are meaningful. This might also clean up some of the differences on Table 3 that are likely unreliable given small cell sizes (e.g., race/ethnicity).
--	--

VERSION 1 – AUTHOR RESPONSE

Reviewer: 1

It is not clear how many PCHUs will be involved in the study. Figure 1 suggested that there will be a total of 54. However, Figures 2 and 3 show an N of 9 per arm. I am wondering if there are multiple waves of PCHUs starting up? Maybe three waves of 18? But this is not clear from the paper.

RESPONSE: It is 54. Figures 2 and 3 had been drawn for the numbers of centres within each country, rather than across the three countries. We have redrawn the figures for the numbers across the three countries and have clarified the numbers in the text.

Clarifying the number of PCHUs involved as well as the timeline of when they will come on board would also help to understand the non-inferiority test of the short clinical package vs. the standard clinical package. I am assuming that this will be tested with the first 9 PCHUs enrolled in the intervention arm and then for future enrolled sites, the standard clinical package arm will be eliminated (if hypothesis of non-inferiority is confirmed) but again this is not clear from the paper.

RESPONSE: Please see response above. We have clarified the numbers of PHCUs in the text, and expanded the description of the non-inferiority test with the numbers of the PHCU.

The authors include a timeline for the project but this is buried in the supplemental attachments, not referenced in the paper, and overly detailed. I suggest the authors create a timeline specific for the protocol paper that shows just key elements such as when sites will be onboarded and when

assessments will take place. This should be presented early on in the paper to orient the reader to the overall project.

RESPONSE: Yes, we have added a timeline figure to the main body of the text and deleted the longer and overly detailed timeline from the supplementary material.

As presented currently, the Data Collection and Instruments section is a list of instruments that are not organized by when the assessments will occur again making it difficult for the reader to gain a coherent understanding of what is occurring when during the project. I would suggest organizing the section by time, for example describing information collected prior to implementation, during provider training, during patient recruitment phase and post-implementation period. This can be tied to the timeline suggested above.

RESPONSE: We have ordered by time and heading, and matched this to the timeline figure added to the main body of the text.

Key informant interviews and observations seem to be part of the Process Evaluation and so should be described under that section.

RESPONSE: We have moved the relevant text to the process evaluation section.

In the "Implementation, mechanisms of impact and context" section, the authors describe analyzing factors within five groups. The five groups are listed in the first paragraph of this section, then repeated under "The five groups will be assessed as follows:" and described in Table 6. Please, keep the order of the constructs consistent across all three of these and include all five constructs in Table. Currently Description of the Intervention and Outcomes are missing from the table.

RESPONSE: We have done so, and added the Description of the Intervention and Outcomes to the Supplement Table 6.

There needs to be some additional clarity on how the patient tally sheets will be collected. Is it going to be a requirement that each provider use these sheets with every patient? Is this the case for all arms? Or is it up to each provider whether or not they use the tally sheet and for whom? Is it providers that agree to be enrolled in the study that are using the tally sheets and those who decline participation are not? I ask these questions because the primary outcome is the portion of patients who are assessed with the AUDIT-C which is part of the tally sheet so it is very important for the reader to understand the parameters around how that data is collected.

RESPONSE: Rather than simply relying on the figures and tables in the paper and the supplement, we have added further descriptive text to the main body of the text to explain how the study works in each of the arms, including a paragraph describing the use of the tally sheets and their collection.

Many acronyms throughout are not spelled out on first use and may not be familiar to all readers, e.g., PHC, NCD, OECD, ODHIN, SBIRT

RESPONSE: We have included a list of abbreviations and acronyms at the beginning of the paper, and have ensured that they are spelt out the first time that they are mentioned in the text.

Reviewer: 2

Abstract:

Include the number of primary care-based clinics that are proposed to be enrolled in and the aims of the study.

RESPONSE: We have done so.

Briefly clarify the type and duration of training that primary care-based clinics will receive. For example, is it training on screening and managing heavy drinking?

RESPONSE: We have done so.

Consider noting that the study is a four-arm (or three-arm) quasi-experimental study to clarify the original design and inform comparisons on primary outcome.

RESPONSE: We have done so.

From the abstract and introduction, it is not clear how comorbid depression will be addressed in the study.

RESPONSE: Yes, this was not very well explained. We have expanded text throughout on how comorbid depression is dealt with.

Introduction:

Several acronyms have not been defined (e.g., OECD, NCD, CAB).

RESPONSE: We have included a list of abbreviations and acronyms at the beginning of the paper and have ensured that they are spelt out the first time that they are mentioned in the text.

Provide a reference for sentence in the introduction that reports “one in three or four patients being eligible for advice in many countries.”

RESPONSE: We have added two references.

The protocol indicates three interventions, including intensity of clinical package and training, training of providers and community integration and support. However, the difference between the two interventions involving training is not clear. Also, consider modifying Hypothesis 2 to note that it is limited to comparator clinics without municipal support.

RESPONSE: We have clarified both points when describing the study design and when mentioning the hypotheses, correcting the compared arms, where needed.

Although the rationale for including comorbid depression is provided, the focus and description of the procedures related to depression are limited.

RESPONSE: Yes, this was not very well explained. We have expanded text throughout on how comorbid depression is dealt with.

Because the term “clinical package” is somewhat vague and was not formally defined in the narrative, consider replacing the terms “clinical package” with “alcohol screening and treatment procedures” and/or formally defining what clinical package means early in the protocol.

RESPONSE: We have opted for formally defining the clinical package when it is first mentioned.

Methods:

Clarify on Figure 1 when the randomization procedure occurs. The narrative suggests the training and screening procedures are randomized. Additionally, the screening and training procedures on Figure 1 are not consistent with what is reported for comparator clinics on Figures 2 and 3.

RESPONSE: Thank you for spotting this. Allocation to the municipal areas is made by the investigators, as it was not possible to randomize the municipal areas as explained in the text. Within the municipal areas, the PHCU have been randomized to the relevant arms. Figure 1 has been redrawn, with relevant explanatory text added to the main body of the text. Figures 2 and 3 had been drawn for the numbers of centres within each country, rather than across the three countries. We

have redrawn the figures for the numbers across the three countries and have clarified the numbers in the text.

Without information on the “clinical packages” presented early in the methods, it is difficult to fully understand the comparison between the standard and short clinical package and related trainings for Hypothesis 3. Essentially, what are the standard vs. short clinical packages and related trainings? For example, the information presented under the heading “Provider-based measurement and assessment of alcohol consumption...” is difficult to follow without defining the differences in the screening/treatment packages.

RESPONSE: The difference between the packages is described in Supplement Table 1. We have added a summary of the differences in the main body of the text when we refer to Supplement Figures 1 and 2. We have ensured that we use the same terminology (standard and short) throughout.

Clarify whether municipal and primary care based clinical leaders are considered key stakeholders for the purpose of key informant interviews.

RESPONSE: They are, and we have added this clarification to the text.

Although referred to in the proposal, little information is provided about community advisory board and user panels, including how they are formed and the purpose they serve.

RESPONSE: We have added text to the main body of the paper.

Given the abundance of statistical comparisons in a large sample, consider using an approach to correct for the number of comparisons. While reviewing the results on the Tables, there are several comparisons of means that are statistically significant but represent clinically insignificant differences. A higher p-value threshold might eliminate some of these differences, which would make it easier to present and highlight the results that are meaningful. This might also clean up some of the differences on Table 3 that are likely unreliable given small cell sizes (e.g., race/ethnicity).

RESPONSE: With apologies, but we wonder if this comment refers to a separate paper that has been reviewed. As our paper is a protocol paper, we do not yet have any results, nor tables of results as mentioned in this comment.

VERSION 2 – REVIEW

REVIEWER	Hildi Hagedorn Center for Care Delivery and Outcomes Research, Minneapolis VA Health Care System, USA
REVIEW RETURNED	04-May-2020

GENERAL COMMENTS	The authors have responded substantively to most of the previous review comments. This is a very large and complicated trial and their revisions have made it easier to understand how all of the pieces fit together. There is one main outstanding issue that remains unclear. This involves how individual providers are recruited for participation in the study and the procedures for encouraging completion of the tally sheets. Will only providers who consent to participate in the study have access to the tally sheets? Will there be monitoring in place to promote completion of the tally sheets for research data? This is important because the primary outcome comes directly off of the tally sheets. If there is a research infrastructure in place to persuade and remind providers
---

	to complete tally sheets on every patient they see this will artificially inflate the primary outcome measure compared to if the providers are consented, go through their designated training and then are left to their own judgment as to when to implement the tally sheets.
--	--

REVIEWER	Eric Hawkins University of Washington, Department of Psychiatry and Behavioral Sciences, Seattle, WA United States
REVIEW RETURNED	02-May-2020

GENERAL COMMENTS	Overall, authors were very responsive to reviewers' concerns. One additional minor edits was identified during this review. 1) pg. 8, 1st para, It appears that the sentence "As evidence suggests that shorter sessions of brief advice are not less effective compared to shorter sessions" should highlight the point that shorter sessions of brief advice are not less effective than longer sessions?
--

VERSION 2 – AUTHOR RESPONSE

Reviewer: 2

1) pg. 8, 1st para, It appears that the sentence "As evidence suggests that shorter sessions of brief advice are not less effective compared to shorter sessions" should highlight the point that shorter sessions of brief advice are not less effective than longer sessions?

RESPONSE: Thank you for spotting this. This was a typing mistake. It has been corrected to: "As evidence suggests that shorter sessions of brief advice are not less effective compared to longer sessions..".

Reviewer: 1

There is one main outstanding issue that remains unclear. This involves how individual providers are recruited for participation in the study and the procedures for encouraging completion of the tally sheets. Will only providers who consent to participate in the study have access to the tally sheets? Will there be monitoring in place to promote completion of the tally sheets for research data? This is important because the primary outcome comes directly off of the tally sheets. If there is a research infrastructure in place to persuade and remind providers to complete tally sheets on every patient they see this will artificially inflate the primary outcome measure compared to if the providers are consented, go through their designated training and then are left to their own judgment as to when to implement the tally sheets.

RESPONSE:

We have changed the text to:

"Within each PHCU, individual providers decide themselves whether or not to participate in the study; those who do sign an informed consent for their participation. Based on the five-country ODHIN study, we estimate that approximately two-fifths of providers will consent to join the study.⁶¹"

During the description of the baseline measurement period, we have added:

“Tally sheets will be delivered to the PHCU to be distributed to the participating providers at the beginning of the one-month baseline measurement period and collected at the end of the period, with no other contact during the period”.

During the description of the 18-month implementation period, we have added:

“Tally sheets will be delivered monthly to each PHCU to distribute to participating providers. Completed tally sheets will be collected at the end of each month. Following training in Arms 2 to 4, and municipal support in Arms 3 to 4, each provider determines use and completion of the tally sheets, with no additional prompting”.

VERSION 3 – REVIEW

REVIEWER	Hildi Hagedorn Center for Care Delivery & Outcomes Research, Minneapolis Veterans Affairs Healthcare System, Minneapolis, Minnesota, USA
REVIEW RETURNED	13-May-2020
GENERAL COMMENTS	The authors have addressed my remaining question. I have no further issues to raise.